# DUAL CONSOLIDATION FOR PRE-TRAINED MODEL-BASED DOMAIN-INCREMENTAL LEARNING

## ABSTRACT

Domain-Incremental Learning (DIL) involves the progressive adaptation of a model to new concepts across different domains. While recent advances in pre-trained models provide a solid foundation for DIL, learning new concepts often results in the catastrophic forgetting of pre-trained knowledge. Specifically, sequential model updates can overwrite both the representation and the classifier with knowledge from the latest domain. Thus, it is crucial to develop a representation and corresponding classifier that accommodate all seen domains throughout the learning process. To this end, we propose DUal ConsolidaTion (DUCT) to unify and consolidate historical knowledge at both the *representation* and *classifier* levels. By merging the backbone of different stages, we create a representation space suitable for multiple domains incrementally. The merged representation serves as a balanced intermediary that captures task-specific features from all seen domains. Additionally, to address the mismatch between consolidated embeddings and the classifier, we introduce an extra classifier consolidation process. Leveraging class-wise semantic information, we estimate the classifier weights of old domains within the latest embedding space. By merging historical and estimated classifiers, we align them with the consolidated embedding space, facilitating incremental classification. Extensive experimental results on four benchmark datasets demonstrate DUCT's state-of-the-art performance.

## 1 INTRODUCTION

Recent years have seen the rise of deep learning, demonstrating its strong potential in real-world applications (He et al., 2016; Deng et al., 2009; Floridi & Chiriatti, 2020). However, in a dynamic and ever-changing world, data often evolves in a stream format (Aggarwal, 2018), necessitating continuous updates with data from *new domains* (Gama et al., 2014). For instance, autonomous vehicles must handle driving tasks across different seasons and weather conditions (Bojarski et al., 2016), and face recognition systems are supposed to recognize users despite the varying lighting conditions (Zhao et al., 2003). Correspondingly, Domain-Incremental Learning (DIL) (van de Ven et al., 2022) addresses this challenge by absorbing new knowledge from new data distributions while preserving existing knowledge. However, sequentially updating the model with new domains often biases the embedding (Yu et al., 2020) and classifier (Zhao et al., 2020; Wu et al., 2019) modules toward the latest domain, leading to catastrophic forgetting (Mermillod et al., 2013; De Lange et al., 2021) of previous knowledge. Therefore, developing methods for learning new domains without forgetting existing knowledge becomes a critical challenge for the machine learning community.

To combat catastrophic forgetting, recent works leverage strong pre-trained models (PTMs)(Han et al., 2021) as the initialization for DIL(Wang et al., 2022d;c;b). PTMs' robust representation abilities provide a powerful starting point, showing promising results in DIL benchmarks. These approaches freeze the pre-trained backbone to preserve existing knowledge and append lightweight modules (Jia et al., 2022; Chen et al., 2022; Lian et al., 2022) to capture new patterns. However, since incoming domains will also overwrite the appended modules, the representations still suffer from forgetting, leading to corresponding bias in the classifier.

In DIL, there are two primary sources of forgetting: the overwriting of *features* and *classifiers*. Continually adjusting the features to capture the latest task biases the representations toward the newest domain. For example, suppose the previous task involves real-world photos while the

subsequent task includes quick-draw images. The features will be overwhelmingly adjusted to highlight non-objective details, which are less helpful in distinguishing the previous domain, thus leading to forgetting. In light of this, an ideal feature space in DIL should fit all domains and equally highlight the domain-specific features for all seen tasks. However, as features become less generalizable due to overwriting, the classifier also becomes biased toward the latest domain. Since the classifier plays the pivotal role of mapping embeddings to classes, using the biased classifier can cause detrimental effects to decision-making. Consequently, it is imperative to consolidate existing knowledge both in the feature and classifier to resist forgetting.

There are two main challenges to achieving this goal. **1)** Building a unified embedding space that suits all tasks continually. Since the training data arrives in a stream format, we cannot access all training instances simultaneously, preventing us from achieving the 'oracle embedding' that favors all seen domains. **2)** Calibrating the prediction of biased classifiers. Even if we achieve a holistic embedding suitable for all tasks, we still cannot build an accurate mapping between features and classes due to the lack of previous instances. Consequently, a classifier consolidation process is needed to align the classifier weights with the changing features continually.

In this paper, we propose DUal ConsolidaTion (DUCT) to address these challenges. To counter feature drift, we introduce a representation merging technique that continuously integrates historical backbones. This merged embedding consolidates task-specific information from all previous domains, providing informative features without forgetting. Consequently, we achieve non-forgettable features incrementally and resist feature-level forgetting. To map updated features to classes, we design a classifier consolidation process that merges historical classifier weights with calibrated weights. By extracting class-wise semantic relationships in the joint space, we develop a classifier transport mechanism that estimates classifiers of previous domain classes in the latest embedding space. Through consolidating features and classifiers in this coordinated way, DUCT balances all seen domains and robustly resists forgetting, achieving competitive performance on various benchmarks.

## 2 RELATED WORK

**Continual Learning/Incremental Learning** is a popular topic in the machine learning field, aiming to incorporate new knowledge within streaming data (De Lange et al., 2021; Masana et al., 2023; Wang et al., 2024). It can be further classified into three subcategories, *i.e.*, task-incremental learning (TIL) (De Lange et al., 2021), class-incremental learning (CIL) (Masana et al., 2023), and domain-incremental learning (DIL) (van de Ven et al., 2022). Among them, TIL and CIL are faced with emerging new classes and are required to learn them without forgetting. By contrast, DIL mainly focuses on the scenario where label space is fixed while incoming data involves new domains (Wang et al., 2022b; Shi & Wang, 2023). In continual learning, typical solutions include using data replay (Bang et al., 2021; Aljundi et al., 2019b; Isele & Cosgun, 2018; Rolnick et al., 2019; Ratcliff, 1990) to review former knowledge, using knowledge distillation (Hinton et al., 2015; Rebuffi et al., 2017; Li & Hoiem, 2017) or parameter regularization (Kirkpatrick et al., 2017; Zenke et al., 2017; Chaudhry et al., 2018; Aljundi et al., 2018; 2019a) to maintain existing knowledge. Recent advances also resort to bias correction (Hou et al., 2019; Zhao et al., 2020; Ahn et al., 2021; Castro et al., 2018; Wu et al., 2019; Belouadah & Popescu, 2019) to rectify the model bias or expand the network structure as data evolves (Rusu et al., 2016; Aljundi et al., 2017; Yan et al., 2021; Wang et al., 2022a; Zhou et al., 2023).

**Domain-Incremental Learning with Pre-Trained Models:** With the rapid development of pre-training techniques, PTMs provide a strong initialization for DIL models to boost the feature generalizability (Smith et al., 2023; Wang et al., 2022c;d;b). Current PTM-based DIL methods mainly resort to visual prompt tuning (Jia et al., 2022) to adjust the pre-trained features. With the pre-trained features frozen, it learns a prompt pool as external knowledge and selects instance-specific prompts to encode task information. L2P (Wang et al., 2022d) utilizes the query-key matching mechanism to select instance-specific prompts, while DualPrompt (Wang et al., 2022c) extends it by learning task-specific and instance-specific prompts jointly. By contrast, S-Prompts (Wang et al., 2022b) learns domain-specific prompts for DIL and retrieves prompts via KNN search. To alleviate the prompt selection cost, CODA-Prompt (Smith et al., 2023) modifies the prompt selection process into an attention-based weighted combination. Apart from prompt tuning, there are also some methods that directly build classifiers upon the pre-trained features (Zhou et al., 2024a; McDonnell et al., 2023).

**Model Merging:** Recent advances in pre-training have made weight interpolation among different models a useful technique in model editing (Ainsworth et al., 2023; Frankle et al., 2020; Ilharco et al., 2023; Matena & Raffel, 2021; Wortsman et al., 2022a;b). Most work focuses on simple averaging of multiple models, aiming to enhance model in terms of its performance on the single task and robustness to out-of-distribution data. In this paper, we consider merging the pre-trained weights and task vector parameters to cultivate multitask representation for all seen domains.

## 3 FROM OLD DOMAINS TO NEW DOMAINS

In this section, we introduce the background information about domain-incremental learning and the baselines for applying pre-trained models to domain-incremental learning.

### 3.1 DOMAIN-INCREMENTAL LEARNING

In domain-incremental learning, the model is faced with the data stream with a sequence of tasks $\left\{\mathcal{D}^1, \mathcal{D}^2, \cdots, \mathcal{D}^B\right\}$, where $\mathcal{D}^b = \{\mathcal{X}_b, \mathcal{Y}_b\}$ is the $b$-th training task, *i.e.*, $\mathcal{X}_b = \{\mathbf{x}_i\}_{i=1}^{n_b}$ and $\mathcal{Y}_b = \{y_i\}_{i=1}^{n_b}$. Each training instance $\mathbf{x}_i \in \mathbb{R}^D$ belongs to class $y_i \in Y$. This paper follows the **exemplar-free** setting (Zhu et al., 2021; Wang et al., 2022d), *i.e.*, during the $b$-th training stage, we can only access data in the current dataset $\mathcal{D}^b$. In DIL, the label space $Y$ is unchanged throughout the learning process, while the distribution of input instance keeps changing from domain to domain, *i.e.*, $p(\mathcal{X}_b) \neq p(\mathcal{X}_{b'})$ for $b \neq b'$. The target of DIL is to fit a model $f(\mathbf{x})$ that can discriminate the classes among any seen domains:

$$f^* = \underset{f \in \mathcal{H}}{\arg\min} \ \mathbb{E}_{(\mathbf{x},y) \sim \mathcal{D}_t^1 \cup \cdots \mathcal{D}_t^b} \mathbb{I}(y \neq f(\mathbf{x})), \tag{1}$$

where $\mathcal{H}$ is the hypothesis space, $\mathbb{I}(\cdot)$ is the indicator function which outputs 1 if the expression holds and 0 otherwise. $\mathcal{D}_t^b$ denotes the data distribution of task $b$. Consequently, DIL models are supposed to classify instances of new domains while not forgetting previous ones.

Following (Wang et al., 2022c;d;b; Smith et al., 2023), we assume a pre-trained Vision Transformer (ViT) (Dosovitskiy et al., 2020) is available as the initialization for $f(\mathbf{x})$. For simplicity, we decouple the network structure into the embedding function $\phi(\cdot) : \mathbb{R}^D \to \mathbb{R}^d$ (*i.e.*, the final `[CLS]` token) and the linear classifier $W \in \mathbb{R}^{d \times |Y|}$. In this way, the output is denoted as $f(\mathbf{x}) = W^\top \phi(\mathbf{x})$, and we utilize a cosine classifier in this paper. The classifier is further decoupled into $W = [\mathbf{w}_1, \mathbf{w}_2, \cdots, \mathbf{w}_{|Y|}]$, where $\mathbf{w}_j$ denotes the classifier weight of the $j$-th class. Although the label space is static during the learning process, data distribution of the same class across different domains can yield significant domain gaps. Hence, we follow existing works (Wang et al., 2022c;d;b; Smith et al., 2023) to expand the classifier $W \in \mathbb{R}^{d \times b|Y|}$ as new tasks emerge, and the final prediction is made by $\left(\arg\max_i \mathbf{w}_i^\top \phi(\mathbf{x})\right) \mathbf{mod} \ |Y|$.

### 3.2 BASELINES IN DOMAIN-INCREMENTAL LEARNING

In domain-incremental learning, a naive solution facing the incoming datasets of new domains is to directly optimize the embedding and classifier:

$$\min_{W \cup \phi} \sum_{(\mathbf{x},y) \in D^b} \ell\left(W^\top \phi(\mathbf{x}), y\right). \tag{2}$$

With pre-trained weights as initialization, Eq. 2 can quickly capture the discriminative features within the new domain. However, since the embedding $\phi(\cdot)$ is kept changing among different domains, it quickly loses the generalization ability to previous domains and suffers from forgetting.

To resist feature-level forgetting, a feasible solution is to freeze the pre-trained weights and append lightweight modules to encode domain-specific knowledge, *e.g.*, prompt (Jia et al., 2022). A representative work L2P (Wang et al., 2022d) organizes a set of prompts as the prompt pool (*i.e.*, **Pool**) and optimizes the model via:

$$\min_{\mathbf{Pool} \cup W} \sum_{(\mathbf{x},y) \in D^b} \ell\left(W^\top \bar{\phi}\left(\mathbf{x}; \mathbf{Pool}\right), y\right) + \mathcal{L}_{\mathbf{Pool}}, \tag{3}$$

where $\bar{\phi}(\mathbf{x}; \mathbf{Pool})$ denotes the prompted feature representation with backbone frozen, and $\mathcal{L}_{\mathbf{Pool}}$ denotes the prompt selection loss (Wang et al., 2022d). Eq. 3 dynamically retrieves instance-specific prompts to adjust the representations and learns the classifier to map the features to corresponding classes. Since the backbone weights are frozen, it can alleviate the representation drift during updating.

**Discussions:** Although Eq. 2 and Eq. 3 adopt different policies for model updating, both of them suffer from the *feature-level* forgetting. Specifically, tuning the model via Eq. 2 can encode task-specific knowledge into the embedding, but also comes with the risk of feature being fully overwritten by new domains since there is no restriction on retaining previous knowledge. Besides, although Eq. 3 freezes the pre-trained weights, the trainable prompts appended may also incur forgetting. Since the prompted feature relies on instance-specific prompt selection, optimizing the prompts for new domains still has the risk of overwriting previous ones. With the features being biased toward the latest domain, the classifier is also adjusted upon it. Consequently, the classifiers of previous tasks are incompatible with the unstable and ever-changing features, resulting in the *classifier-level* forgetting.

## 4    DUCT: DUAL CONSOLIDATION FOR DOMAIN-INCREMENTAL LEARNING

Observing the risk of feature-level and classifier-level overwriting, we aim to resist forgetting in DIL from two aspects. Firstly, alleviating feature-level forgetting requires obtaining a unified embedding space at a low cost. Since sequentially updating the backbone or prompts will both result in forgetting, a proper solution is needed to make full use of all historical features and consolidate them into a unified one. Secondly, as the representation changes from task to task, the mismatch between classifier and embedding becomes more severe as the data evolves. Consequently, a classifier consolidation step is needed to calibrate them to be compatible with the embeddings.

In the following sections, we introduce the learning paradigm for representation consolidation and classifier consolidation. Lastly, we provide detailed guidelines for training and inference.

### 4.1    REPRESENTATION CONSOLIDATION

Observing the challenge of striking a balance between all seen domains, we need to opt for another feature updating policy to avoid sequential overwriting and resist forgetting. Let us assume an ideal scenario where we can separately train each model for each incoming domain, obtaining a set of models $\{\phi_1(\cdot), W_1\}, \cdots, \{\phi_B(\cdot), W_B\}$. Those models are obtained by optimizing the same PTM via Eq. 2, thus having the discriminability for each specific domain and can be seen as the 'domain expert.' In an oversimplified scenario where we know which domain the instance is from, we can directly utilize the corresponding expert for prediction. However, since such auxiliary information is unavailable in DIL, we need to obtain a *omnipotent* embedding space that suits all domains. In this way, there is no need to decide which embedding to use since the omnipotent embedding is strong enough to capture all domain-specific features.

Correspondingly, we take inspiration from the model merging community (Ilharco et al., 2023; Ramé et al., 2023) that an ideal model for multiple domains can be achieved by combining multiple *task vectors*. We denote the relative change of the embedding function as $\delta_{\phi_i} = \phi_i - \phi_0$, where $\phi_0$ is the weight of the pre-trained model. Hence, $\delta_{\phi_i}$ represents the relative change of the weights in the $i$-th domain, and we can build the unified embedding via:

$$\phi_i^m = \phi_0 + \alpha_\phi \sum_i \delta_{\phi_i}, \tag{4}$$

which combines the pre-trained weight with every task vector. Since all the fine-tuned embeddings start from the same pre-trained weight, (Ilharco et al., 2023; Zhang & Bottou, 2023) verify the task vectors share low similarity due to the semantic gap between different domains. Consequently, adding such weights enables the merged model to highlight all task-specific features, and Eq. 4 provides a feasible way to obtain the universal feature representation that suits all seen domains.

**Representation consolidation with task similarity**: Although Eq. 4 provides a simple way to consolidate multiple task vectors to build a unified embedding, it still lacks consideration in measuring task-wise similarities. For example, if two distinct domains are more similar, highlighting those

Figure 1: Illustration of DUCT. **Top**: Representation consolidation. We utilize the pre-trained model as initialization and optimize it for each domain, obtaining the task vectors. Afterward, we combine the pre-trained model and all seen task vectors to build the unified embedding space. **Bottom**: Classifier consolidation. To align the classifiers with consolidated features, we design the new classifier retraining and old classifier transport to consolidate classifiers. Class-wise semantic information is utilized in classifier transport.

features would be more helpful for recognizing classes of corresponding domains. Hence, we introduce task-similarity ($\mathbf{Sim}_{0,i}$) into the representation consolidation process and replace Eq. 4 into:

$$\phi_i^m = \phi_0 + \alpha_\phi \sum_i \mathbf{Sim}_{0,i} \delta_{\phi_i} \, , \tag{5}$$

Specifically, since the merged embedding is built upon pre-trained weights and task vectors, we consider the similarity of pre-trained model $\phi_0$ and subsequent models $\phi_i$ to adjust the merging process. A naive way to is to directly measure the cosine similarity between $\phi_0$ and $\phi_i$. However, since pre-trained weights are with millions of dimensions, the similarity in such space is ineffective due to curse of dimensionality. To this end, we utilize the current training task as an indicator for task similarity. Specifically, for every class in $\mathcal{D}^b$, we utilize the backbone $\phi_i$ to extract class centers in its embedding space:

$$\boldsymbol{c}_p^i = \sum_{j=1}^{|\mathcal{D}^b|} \mathbb{I}(y_j = p)\phi_i(\mathbf{x}_j)/\sum_{j=1}^{|\mathcal{D}^b|} \mathbb{I}(y_j = p) \, . \tag{6}$$

Afterward, we can obtain two sets of class centers in the embedding space, *i.e.*, $\boldsymbol{C}^0 = [\boldsymbol{c}_1^0, \boldsymbol{c}_2^0, \cdots, \boldsymbol{c}_{|Y|}^0]$, $\boldsymbol{C}^i = [\boldsymbol{c}_1^i, \boldsymbol{c}_2^i, \cdots, \boldsymbol{c}_{|Y|}^i]$. Since those class centers are representative points in the embedding space, we calculate the pair-wise class similarity to indicate task similarities:

$$\mathbf{Sim}_{0,i} = \frac{1}{|Y|} \sum_{j=1}^{|Y|} \text{sim}(\boldsymbol{c}_j^0, \boldsymbol{c}_j^i) \, , \tag{7}$$

where we utilize cosine similarity to calculate center-wise similarity $\text{sim}(\cdot, \cdot)$.

**Effect of representation consolidation**: Figure 1(top) visualizes the representation consolidation process, where the merged backbone can unify multiple domains. Through Eq. 5, we are able to build the joint embedding space that suits all tasks by combining them in the parameter space. Since tasks in DIL emerge one-by-one, the representation consolidation process can also be made *incrementally*, *i.e.*, $\phi_i^m = \phi_{i-1}^m + \alpha_\phi \mathbf{Sim}_{0,i} \delta_{\phi_i}$. In this way, we can get rid of the extensive memory cost and only keep at most two backbones in memory. In each training stage, we first fine-tune the model via Eq. 2 and conduct representation consolidation to aggregate the representations. In this way, we obtain a unified embedding space across all seen tasks.

## 4.2 CLASSIFIER CONSOLIDATION

In Eq. 5, we design the feature consolidation process, which aggregates multiple fine-tuned models by accumulating task vectors. However, a fatal problem still exists, *i.e.*, there is a *mismatch* between the classifiers and the consolidated embeddings. Since the classifiers are optimized to match the embeddings to the corresponding class, the matching degree drastically decays as the backbone is replaced with another one. Recalling the background information in Section 3.1 that we utilize an independent classifier for each domain, we need to design the extra classifier consolidation process to align the classifiers to the embedding space. In each training stage, we denote the classifier for previous domains as 'old classifier' ($W_o$) and the classifier for the current domain as 'new classifier' ($W_n$). We then discuss how to align them with the consolidated features.

**New Classifier Retraining**: In each training stage, we have the training data $\mathcal{D}^b$ in hand. Consequently, it is intuitive to coordinate the new classifier with the consolidated features via:

$$\min_{W_n} \sum_{(\mathbf{x},y)\in D^b} \ell\left(W_n^\top \bar{\phi}_i^m(\mathbf{x}), y\right) , \tag{8}$$

where the consolidated feature $\bar{\phi}_i^m(\cdot)$ is frozen to resist further mismatch, and we can align the new classifier $W_n$ to match the features with Eq. 8.

**Old Classifier Transport**: Eq. 8 aligns the new classifier to the latest embedding space. However, a fatal problem still exists, *i.e.*, the old classifier is also incompatible with the merged embedding space. Consequently, the previous knowledge of old domains shall be forgotten in the incremental learning process. If we have plenty of training instances of previous domains, a similar calibration process can be done as in Eq. 8. However, due to the exemplar-free restrictions, we cannot save any previous instances in the memory. Hence, we need to find another way to estimate the relative calibration weight for the old classifier. For example, if we have the retrained classifier to classify a 'lion' in the *clip art* style, it can be mostly reused to classify the lion in the *real photo* style. We call such a relationship 'semantic information,' and the goal is to utilize such information to assist old classifier alignment. We denote the calibrated classifier using semantic information as $\hat{W}_o = \mathcal{T}(W_n, \mathbf{S})$, *i.e.*, the estimated classifier is obtained by transforming the new classifier using semantic information $\mathbf{S}$. Hence, there remain two core problems to be solved: **1)** How to define the transformation function $\mathcal{T}$? **2)** How to define the semantic information $\mathbf{S}$?

**Implementing $\mathcal{T}$ via Optimal Transport**: The function $\mathcal{T}$ encodes the correlation between the set of features among two tasks. Considering that the weight matrix of a linear layer reveals the relative relationship among feature-class pairs and the final logit is the aggregation of all features, we can design the *recombination* of exiting classifiers with a linear mapping $T \in \mathbb{R}^{\beta \times \gamma}$. $T$ encodes the cross-task correlation between the current and the previous domains, where larger values in $T$ denote higher class-wise similarity. Since the decision is made by matching the classifier with corresponding features, we can utilize and recombine the weights of similar classes in the current domain to obtain those of previous domains. For example, important features that discriminate lions in the clip art style should be assigned higher coefficients to help classify lions in the photo style and vice versa.

We denote $\boldsymbol{\mu}_1 \in \Delta_\beta$ and $\boldsymbol{\mu}_2 \in \Delta_\gamma$, where $\Delta_d = \left\{\boldsymbol{\mu} : \boldsymbol{\mu} \in \mathbb{R}_+^d, \boldsymbol{\mu}^\top \mathbf{1} = 1\right\}$ is the $d$-dimensional simplex. $\boldsymbol{\mu}_1$ and $\boldsymbol{\mu}_2$ denote the importance of each class across domains, and we set them as uniform distribution. To map a set of classes to another between old and new domains, a cost matrix $Q \in \mathbb{R}_+^{\beta \times \gamma}$ is further introduced to guide the transition. The larger weight of $Q_{i,j}$ indicates we need to pay more cost when reusing the classifier of $i$-th class to assist the $j$-th class. Consequently, the matrix $T$ can be formulated as the coupling of two distributions, aiming to connect classes from different domains at the lowest transportation cost. Hence, $T$ can be obtained via minimizing:

$$\min_T \langle T, Q \rangle \quad \text{s.t. } T\mathbf{1} = \boldsymbol{\mu}_1, T^\top \mathbf{1} = \boldsymbol{\mu}_2, T \geq 0 , \tag{9}$$

which is the Kantorovich formulation of optimal transport (OT) (Villani, 2008; Peyré et al., 2019). Optimizing Eq. 9 enables us to show how to move the probability mass across domains with low cost. Since the target is to reuse the retrained classifier $W_n$ to adjust previous classifier $W_o$, we set $\boldsymbol{\mu}_1 \in \Delta_{|Y|}$ and $\boldsymbol{\mu}_2 \in \Delta_{|Y_o|}$ with uniform class marginals, where $|Y_o|$ denotes the number of classes in $W_o$. Solving Eq. 9 produces the alignment between different domains, and we can apply $T$ to the new classifier to obtain the estimated old classifier, *i.e.*, $\hat{W}_o = W_n T$.

**Defining the Transportation Cost with Semantic Information**: Solving Eq. 9 requires a proper definition of the cross-domain cost, *i.e.*, $Q$. The higher cost indicates it is less effective to transport the classifier to the target class and vice versa. To this end, we design a simple yet effective way to measure such class-wise information. Specifically, before the training of each stage, we utilize the pretrained backbone $\phi_0$ to extract class centers in its embedding space via Eq. 6, *i.e.*, $[\boldsymbol{c}_1^0, \boldsymbol{c}_2^0, \cdots, \boldsymbol{c}_{|Y|}^0]$. These class centers indicate the most representative embedding of the corresponding classes. Since $\phi_0$ is optimized with extensive datasets, we rely on its generalizability to reflect task-wise similarity. If two classes are similar, their embeddings should also be situated near each other. Consequently, we calculate the Euclidean distance between class centers as the transportation cost, *i.e.*, $Q_{i,j} = \left\|\boldsymbol{c}_i^0 - \boldsymbol{c}_j^0\right\|_2^2$. Here classes $i$ and $j$ are from different domains. With the pair-wise transportation cost,

Table 1: Average and last performance of different methods among five task orders. The best performance is shown in bold. All methods are implemented with ViT-B/16 IN1K. Methods with † indicate implemented with exemplars (10 per class).

| Method | Office-Home | | DomainNet | | CORe50 | | CDDB | |
|---|---|---|---|---|---|---|---|---|
| | $\bar{\mathcal{A}}$ | $\mathcal{A}_B$ | $\bar{\mathcal{A}}$ | $\mathcal{A}_B$ | $\bar{\mathcal{A}}$ | $\mathcal{A}_B$ | $\bar{\mathcal{A}}$ | $\mathcal{A}_B$ |
| Finetune | $78.32_{\pm3.28}$ | $76.16_{\pm1.39}$ | $28.17_{\pm6.47}$ | $38.82_{\pm7.65}$ | $75.44_{\pm1.68}$ | $76.19_{\pm2.36}$ | $52.08_{\pm1.35}$ | $50.11_{\pm1.62}$ |
| Replay† (Ratcliff, 1990) | $84.23_{\pm2.31}$ | $83.75_{\pm0.68}$ | $64.78_{\pm2.98}$ | $61.16_{\pm1.19}$ | $85.56_{\pm0.38}$ | $92.21_{\pm0.63}$ | $66.91_{\pm18.0}$ | $63.21_{\pm11.6}$ |
| iCaRL† (Rebuffi et al., 2017) | $81.66_{\pm2.43}$ | $81.11_{\pm1.23}$ | $59.89_{\pm2.86}$ | $57.46_{\pm2.31}$ | $74.43_{\pm3.18}$ | $79.86_{\pm2.96}$ | $68.43_{\pm18.7}$ | $70.50_{\pm16.5}$ |
| MEMO† (Zhou et al., 2023) | $71.18_{\pm2.76}$ | $63.09_{\pm1.80}$ | $61.92_{\pm5.39}$ | $58.41_{\pm3.20}$ | $64.80_{\pm3.16}$ | $68.24_{\pm2.41}$ | $60.87_{\pm13.4}$ | $58.09_{\pm11.7}$ |
| SimpleCIL (Zhou et al., 2024a) | $75.69_{\pm5.03}$ | $75.72_{\pm0.00}$ | $42.95_{\pm4.84}$ | $44.08_{\pm0.00}$ | $70.92_{\pm0.74}$ | $74.80_{\pm0.00}$ | $60.80_{\pm4.49}$ | $63.40_{\pm0.00}$ |
| L2P (Wang et al., 2022d) | $79.72_{\pm4.19}$ | $80.03_{\pm1.29}$ | $50.45_{\pm4.10}$ | $48.72_{\pm2.83}$ | $83.57_{\pm0.35}$ | $87.87_{\pm0.51}$ | $67.33_{\pm5.98}$ | $64.45_{\pm5.83}$ |
| DualPrompt (Wang et al., 2022c) | $80.20_{\pm3.81}$ | $80.85_{\pm0.14}$ | $52.28_{\pm3.35}$ | $50.46_{\pm3.17}$ | $84.53_{\pm0.89}$ | $87.27_{\pm1.06}$ | $68.33_{\pm7.52}$ | $71.41_{\pm1.24}$ |
| CODA-Prompt (Smith et al., 2023) | $84.70_{\pm2.94}$ | $85.07_{\pm0.34}$ | $59.85_{\pm4.49}$ | $59.99_{\pm0.88}$ | $87.92_{\pm0.41}$ | $91.57_{\pm0.69}$ | $69.19_{\pm6.11}$ | $74.18_{\pm1.33}$ |
| EASE (Zhou et al., 2024b) | $81.16_{\pm3.52}$ | $76.33_{\pm2.16}$ | $50.50_{\pm2.27}$ | $43.72_{\pm1.70}$ | $86.30_{\pm0.04}$ | $87.02_{\pm1.21}$ | $67.78_{\pm2.44}$ | $64.96_{\pm8.36}$ |
| RanPAC (McDonnell et al., 2023) | $82.30_{\pm3.34}$ | $82.28_{\pm0.00}$ | $55.20_{\pm3.93}$ | $54.80_{\pm0.36}$ | $79.16_{\pm0.55}$ | $81.38_{\pm0.12}$ | $78.92_{\pm4.85}$ | $80.48_{\pm0.68}$ |
| S-iPrompt (Wang et al., 2022b) | $81.50_{\pm3.17}$ | $80.51_{\pm0.21}$ | $61.16_{\pm4.57}$ | $60.46_{\pm0.90}$ | $81.95_{\pm0.57}$ | $83.38_{\pm0.70}$ | $68.51_{\pm7.20}$ | $72.76_{\pm0.43}$ |
| DUCT | $\mathbf{86.27}_{\pm2.95}$ | $\mathbf{86.91}_{\pm0.06}$ | $\mathbf{67.16}_{\pm3.75}$ | $\mathbf{67.01}_{\pm1.35}$ | $\mathbf{91.95}_{\pm0.15}$ | $\mathbf{94.47}_{\pm0.33}$ | $\mathbf{84.14}_{\pm4.37}$ | $\mathbf{85.10}_{\pm0.52}$ |

we are able to optimize Eq. 9 for the transportation plan $T$. Afterward, we utilize the interpolation between old classifiers and the transported classifier for consolidation:

$$W_o^m = (1 - \alpha_W)W_o + \alpha_W \hat{W}_o = (1 - \alpha_W)W_o + \alpha_W W_n T. \tag{10}$$

**Effect of classifier consolidation**: We visualize the classifier consolidation process in Figure 1 (bottom). The classifier consolidation process takes two steps, *i.e.*, new classifier retraining and old classifier transport. The first step is designed to align the new classifiers with the unified embedding, while the second step aims to recombine the old classifiers with the calibrated classifier. In this way, we obtain a compatible classifier with the consolidated features, thus favoring the recognition in the unified embedding space.

### 4.3 SUMMARY OF DUCT

We summarize the training steps in Algorithm 1. Specifically, facing a new training task, we extract the class centers with $\phi_0$. Afterward, we optimize the model and separately consolidate the features and classifiers. The consolidated model ($[W_o^m; W_n]^\top \phi_i^m$) is then utilized for testing. We utilize Sinkhorn's algorithm (Sinkhorn & Knopp, 1967) to solve the OT problem in Eq. 9.

**Discussions on memory cost**: During the training process, we need to keep two models in the memory, *i.e.*, the historical consolidated backbone $\phi_i^m$ and the online backbone $\phi_i$ for learning the current task. However, after the learning process of each domain,

---

**Algorithm 1** DUCT for DIL

**Input**: Incremental datasets: $\left\{ \mathcal{D}^1, \mathcal{D}^2, \cdots, \mathcal{D}^B \right\}$,
Pre-trained embedding: $\phi_0(\mathbf{x})$;
**Output**: Incrementally trained model;
1: **for** $b = 1, 2 \cdots, B$ **do**
2:    Get the incremental training set $\mathcal{D}^b$;
3:    Extract class centers via Eq. 6;
4:    Optimize the model via Eq. 2 ;
5:    Consolidate representations via Eq. 5;
6:    Retrain new classifiers via Eq. 8;
7:    Solve OT via Eq. 9;
8:    Consolidate old classifiers via Eq. 10;
  **return** the updated model;

---

the current running model is merged into $\phi_i^m$ (Eq. 5) without needing to be kept in memory. During inference, we utilize the consolidated backbone $\phi_i^m$ and classifiers $[W_o^m; W_n]$ for classification, which is the same as a single backbone. Consequently, DUCT can be fairly compared with others.

## 5 EXPERIMENT

In this section, we conduct experiments on benchmark datasets to compare DUCT to existing state-of-the-art methods. We also provide ablation studies on the components and parameter robustness to analyze the effect of different modules. Visualizations of the embedding space also verify the effectiveness of our proposed method.

### 5.1 EXPERIMENTAL SETUP

**Dataset**: Following the benchmark settings (Wang et al., 2022c;d;b; Smith et al., 2023) in PTM-based domain-incremental learning, we evaluate the performance on **Office-Home** (Venkateswara et al.,

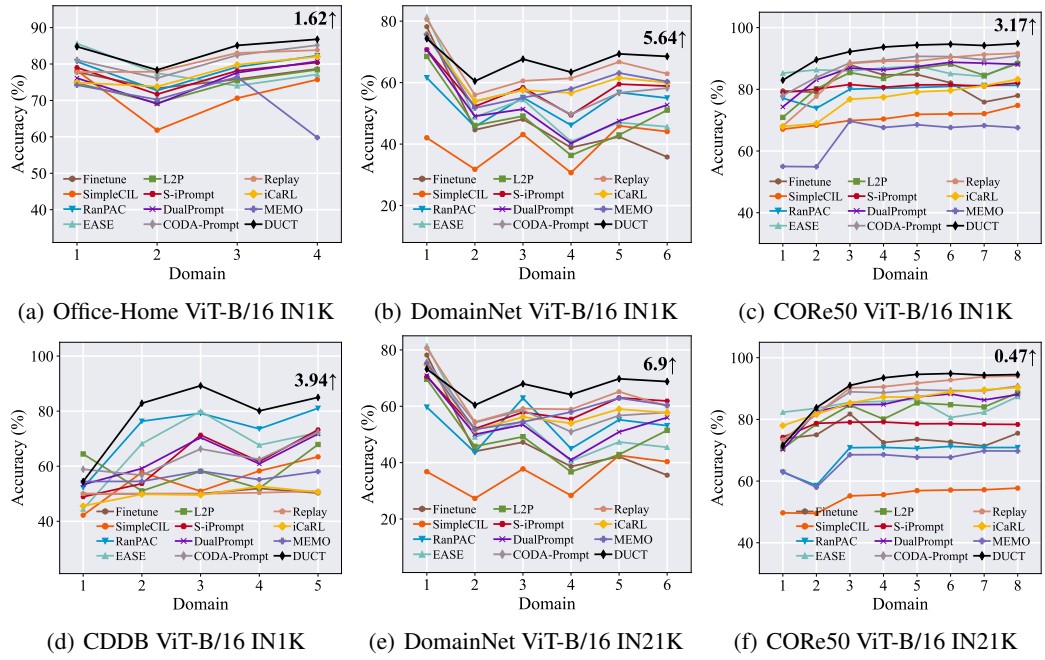

(a) Office-Home ViT-B/16 IN1K     (b) DomainNet ViT-B/16 IN1K     (c) CORe50 ViT-B/16 IN1K

(d) CDDB ViT-B/16 IN1K     (e) DomainNet ViT-B/16 IN21K     (f) CORe50 ViT-B/16 IN21K

Figure 2: Incremental performance of different methods with the same pre-trained model. We report the performance gap after the last incremental stage between DUCT and the runner-up method at the end of the line.

2017), **DomainNet** (Peng et al., 2019), **CORe50** (Lomonaco & Maltoni, 2017), and **CDDB-Hard** (Li et al., 2023). Specifically, Office-Home has four domains, DomainNet has six domains, CORe50 has 11 domains, and CDDB-Hard has five domains. We consider five task orders to shuffle these domains in the DIL setting for a holistic evaluation and report the details in Section B.2.

**Comparison methods:** In the comparison, we consider two types of methods, including exemplar-based methods, *i.e.*, Replay (Ratcliff, 1990), iCaRL (Rebuffi et al., 2017), MEMO (Zhou et al., 2023), and SOTA exemplar-free DIL methods, *i.e.*, SimpleCIL (Zhou et al., 2024a), L2P (Wang et al., 2022d), DualPrompt (Wang et al., 2022c), CODA-Prompt (Smith et al., 2023), EASE (Zhou et al., 2024b), RanPAC (McDonnell et al., 2023), and S-iPrompt (Wang et al., 2022b). We utilize the **same pre-trained backbone for all compared methods.**

**Implementation details:** We deploy the experiments using PyTorch (Paszke et al., 2019) on NVIDIA 4090. Following (Wang et al., 2022d; Zhou et al., 2024b), we consider two typical pre-trained weights, *i.e.*, ViT-B/16-IN21K and ViT-B/16-IN1K. Both are pre-trained with ImageNet21K (Russakovsky et al., 2015), while the latter is further finetuned on ImageNet1K. We optimize DUCT using SGD optimizer with a batch size of 128 for 15 epochs. The learning rate is set to $0.001$. We select 10 exemplars per class for exemplar-based methods using herding (Welling, 2009) algorithm. In DUCT, we set the consolidation parameter $\alpha_\phi = 0.5, \alpha_w = 0.5$. The code will be made publicly available upon acceptance.

**Performance Measure:** Following (Wang et al., 2022d;b), we denote the accuracy among all seen domains after learning the $b$-th task as $\mathcal{A}_b$, we mainly consider $\mathcal{A}_B$ (the performance after learning the last stage) and $\bar{\mathcal{A}} = \frac{1}{B} \sum_{b=1}^{B} \mathcal{A}_b$ (the average performance among all incremental stages) for comparison. We also consider the forgetting measure (Chaudhry et al., 2018) to measure the relative forgetting degree in DIL.

## 5.2 BENCHMARK COMPARISON

We report the results on four benchmark datasets in Figure 2. As we can infer from these results, DUCT consistently outperforms other compared methods by $1 \sim 7\%$ in the final accuracy. It must be noted that the differences of the starting point are due to the different tuning techniques, which cannot be directly aligned since the number of trainable parameters are different in those methods.

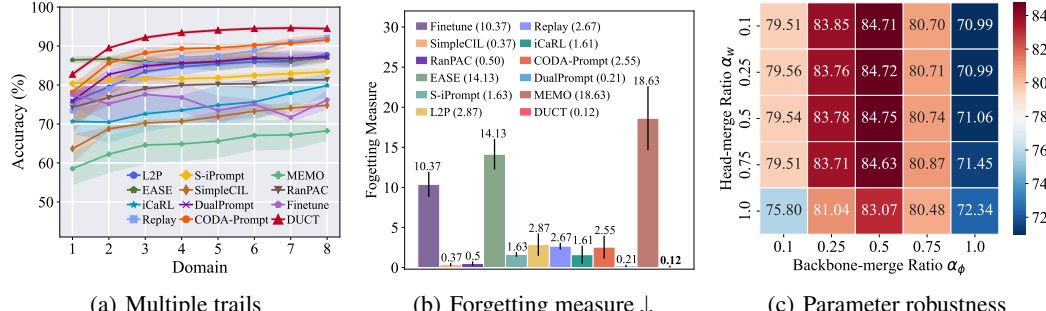

| (a) Multiple trails | (b) Forgetting measure ↓ | (c) Parameter robustness |
|---|---|---|

Figure 3: Further analysis on multiple task orders, forgetting measure, and parameter robustness. **(a):** Incremental performance of different methods on CORe50 with five task orders. The shadow indicates standard deviation. **(b):** Forgetting measure (**lower is better**) of different methods on CDDB dataset among five task orders. DUCT shows the least forgetting among all methods. **(c):** Average incremental performance with change of the consolidation ratios.

Specifically, sequentially finetuning the model suffers from catastrophic forgetting, even starting with the pre-trained model. To compensate for the forgetting phenomena, we observe that several exemplar-based methods (Replay, iCaRL, and MEMO) show stable improvements to the baseline. However, since some of these works are designed for the class-incremental learning scenario, the expansion or distillation target may not be optimal for the current setting. We then compare DUCT to the methods specially designed for PTMs and find prompt-based methods (L2P, DualPrompt, CODA-Prompt, and S-iPrompt) yield inferior performance to ours. We also observe that DUCT is compatible with various pre-trained weights and shows stable improvements with ViT-B/16 IN1K (a∼d) and IN21K (e∼f).

Besides, we also report the incremental performance among five task orders (average and standard deviation) in Table 1 and Figure 3(a). As we can infer from the table, DUCT works robustly among different task orders, showing stable improvements against other state-of-the-art methods.

## 5.3 FURTHER ANALYSIS

**Forgetting Measure**: Figure 2 and Table 1 mainly focus on the accuracy measure, which utilizes all seen domains to measure the relative performance. Apart from the accuracy measure, we also follow (Wang et al., 2022b;d) to utilize the forgetting measure, which captures the stability of existing knowledge among previous domains. As shown in Figure 3(b), we report the forgetting measure among all compared methods on the CDDB dataset. As we can infer from the figure, several methods suffer from severe forgetting, *e.g.*, Finetune and MEMO, indicating that they are unsuitable for the domain-incremental learning scenario. We also observe that prompt-based methods freeze the backbone to resist feature drift, which tackles the forgetting phenomena. However, DUCT still shows the least forgetting (*i.e.*, 0.12) among all compared methods, indicating its strong performance.

**Parameter Robustness:** There are two major consolidation steps in DUCT, *i.e.*, the representation consolidation in Eq. 5 and the classifier consolidation in Eq. 10. These consolidation steps include the merging parameters between a set of models/classifiers, *i.e.*, the backbone merge ratio $\alpha_\phi$ and the classifier merge ratio $\alpha_W$. In this section, we conduct ablations on the choice of these parameters to investigate the robustness of DUCT with change of them. Specifically, we choose $\alpha_\phi$ and $\alpha_W$ among $\{0.1, 0.25, 0.5, 0.75, 1.0\}$, resulting in 25 parameter combinations. We conduct experiments with these parameter combinations on the Office-Home dataset and show the average performance in Figure 3(c). As we can infer from the figure, DUCT is generally robust to parameter changes. Besides, since the merge parameters are designed to trade-off between old and new knowledge, we find a small value (*e.g.*, 0.1) or large value (*e.g.*, 1.0) works poorly. The poor performance of these marginal parameters also indicates the importance of the merging process since a small value or large value equals to ignoring part of the important information during consolidation. By contrast, both of them prefer a mild value, *i.e.*, choosing them around $\{0.25, 0.5\}$ shows better performance. Hence, we suggest setting $\alpha_\phi = 0.5, \alpha_w = 0.5$ as the default setting.

**Compositional Components:** In this section, we conduct experiments to investigate the importance of each module in DUCT on CDDB dataset. As shown in Table 2, 'Baseline' denotes directly freezing

| Variations | $\bar{\mathcal{A}}$ | $\mathcal{A}_B$ |
|---|---|---|
| Baseline | 66.56 | 63.40 |
| Variation 1 | 80.36 | 74.17 |
| Variation 2 | 81.41 | 76.82 |
| Variation 3 | 85.42 | 80.31 |
| DUCT | **87.74** | **82.35** |

Table 2: Ablation study on different modules in DUCT.

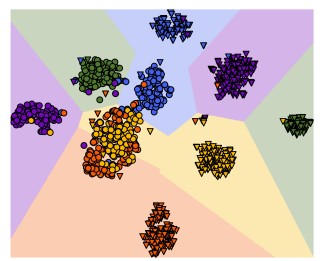

Figure 4: Before DUCT.

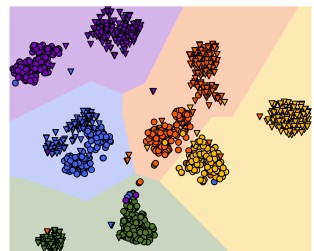

Figure 5: After DUCT.

Figure 6: Visualizations of the embedding space via t-SNE (Van der Maaten & Hinton, 2008) on DomainNet. We visualize two incremental stages and utilize dots to represent the first domain and triangles for the second domain. The shadow region denotes the decision boundaries.

the embedding and extracting class centers as the classifier, which shows poor performance due to the domain gap to the downstream domains. We then utilize Eq. 4 to sequentially merge the backbone to consolidate the representations and denote it as **Variation 1**. It shows that adding the representation consolidation process drastically improves the performance, indicating the superiority of such unified representation in DIL. However, when switching Eq. 4 to Eq. 5, we find **Variation 2** further improves the performance, indicating that task similarity is helpful in obtaining a universal embedding space. Afterward, we combine it with the classifier retraining process in Eq. 8, which is denoted as **Variation 3**. It shows that the mismatch between consolidated features and the classifiers weakens the performance, and aligning the new classifiers to the embedding helps DIL. When comparing DUCT to Variation 3, we find the old classifier transport is also vital to recover former knowledge and resist forgetting, which improves the final accuracy by 2%. Ablation studies verify the efficacy of different modules in DUCT.

**Visualizations:** In this section, we visualize the embedding space to show the effectiveness of DUCT on the DomainNet dataset. We consider a two-stage domain-incremental learning scenario, each containing five classes, and utilize t-SNE (Van der Maaten & Hinton, 2008) to visualize the representation space before and after DUCT. We use dots to represent the classes in the first domain and triangles to represent classes in the second domain. We utilize the shadow regions to denote the decision boundary obtained by the classifier weights to represent different classes. As shown in Figure 4, although the embedding space before DUCT shows competitive performance, there are two major flaws: 1) There exists the *confusion region* between yellow and red dots, indicating that previous classes are partially forgotten as data evolves. 2) The embedding space of the same class is still located in different regions, *e.g.*, the purple and green classes, which is not ideal for domain-incremental learning. However, when applying DUCT to consolidate the features and classifiers, the above problems are addressed in Figure 5. Specifically, since the consolidated embedding suits all tasks, the forgetting of previous domains is alleviated. Besides, the unified embedding space adequately places the same class of different domains together, favoring the final inference.

## 6 CONCLUSION

Domain-incremental learning is a desired ability for real-world learning systems to obtain new knowledge. In this paper, we propose dual consolidation (DUCT) to achieve this goal. Since the forgetting in DIL occurs in two aspects, *i.e.*, the embedding and the classifier, we separately design the consolidation technique to handle them. Specifically, we consider merging the pre-trained model with domain-specific task vectors to achieve the unified embedding. To compensate for the mismatch between consolidated features and classifiers, we design the classifier consolidation process by introducing semantic-guided transport. Extensive experiments validate the effectiveness of DUCT.

**Limitations:** Possible limitations include the utilization of PTMs since feature consolidation relies on generalizable initialization. Future work includes extending DUCT to non-PTM scenarios.

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

# Appendix

In the main paper, we present a method to prevent forgetting in domain-incremental learning through representation and classifier consolidation. The supplementary material provides additional details on the experimental results mentioned in the main paper, along with extra empirical evaluations and discussions. The organization of the supplementary material is as follows:

- Section A presents additional experimental results, including the running time comparison, detailed performance on different task orders, and results with other pre-trained weights.

- Section B introduces the details about the datasets adopted in the main paper, including the number of tasks and images and differently-ordered sequences of tasks.

- Section C introduces the compared methods adopted in the main paper.

Table 3: Average and last performance of different methods with the 1st task order in Section B.2. The best performance is shown in bold. All methods are implemented with ViT-B/16 IN1K. Methods with † indicate implementations with exemplars (10 per class).

| Method | Office-Home | | DomainNet | | Core50 | | CDDB | |
|---|---|---|---|---|---|---|---|---|
| | $\bar{\mathcal{A}}$ | $\mathcal{A}_B$ | $\bar{\mathcal{A}}$ | $\mathcal{A}_B$ | $\bar{\mathcal{A}}$ | $\mathcal{A}_B$ | $\bar{\mathcal{A}}$ | $\mathcal{A}_B$ |
| Finetune | 73.48 | 76.23 | 47.99 | 35.80 | 74.44 | 72.18 | 50.43 | 50.23 |
| Replay† | 80.62 | 83.80 | 64.68 | 62.88 | 85.71 | 91.66 | 50.26 | 50.91 |
| iCaRL† (Rebuffi et al., 2017) | 77.63 | 82.03 | 60.25 | 59.05 | 76.79 | 81.60 | 49.63 | 49.77 |
| MEMO† (Zhou et al., 2023) | 70.17 | 59.80 | 60.65 | 60.33 | 64.89 | 67.55 | 53.00 | 53.67 |
| SimpleCIL (Zhou et al., 2024a) | 71.60 | 75.72 | 39.61 | 44.08 | 70.81 | 74.80 | 54.52 | 63.40 |
| L2P (Wang et al., 2022d) | 74.49 | 78.44 | 49.00 | 51.07 | 83.47 | 88.33 | 58.69 | 67.89 |
| DualPrompt (Wang et al., 2022c) | 75.88 | 80.72 | 51.92 | 52.73 | 85.42 | 88.09 | 63.10 | 71.72 |
| CODA-Prompt (Smith et al., 2023) | 81.20 | 85.17 | 58.49 | 58.21 | 87.70 | 90.85 | 63.33 | 72.28 |
| EASE (Zhou et al., 2024b) | 78.56 | 77.15 | 53.04 | 45.63 | 86.28 | 88.98 | 66.47 | 72.18 |
| RanPAC (McDonnell et al., 2023) | 78.72 | 82.28 | 53.48 | 54.97 | 79.44 | 81.32 | 72.47 | 81.04 |
| S-iPrompt (Wang et al., 2022b) | 77.32 | 80.42 | 58.21 | 58.88 | 80.91 | 82.09 | 61.75 | 73.23 |
| DUCT | **83.76** | **86.79** | **67.28** | **68.52** | **92.06** | **94.83** | **78.32** | **84.98** |

Table 4: Average and last performance of different methods with the 2rd task order in Section B.2. The best performance is shown in bold. All methods are implemented with ViT-B/16 IN1K. Methods with † indicate implementations with exemplars (10 per class).

| Method | Office-Home | | DomainNet | | Core50 | | CDDB | |
|---|---|---|---|---|---|---|---|---|
| | $\bar{\mathcal{A}}$ | $\mathcal{A}_B$ | $\bar{\mathcal{A}}$ | $\mathcal{A}_B$ | $\bar{\mathcal{A}}$ | $\mathcal{A}_B$ | $\bar{\mathcal{A}}$ | $\mathcal{A}_B$ |
| Finetune | 76.50 | 74.37 | 36.94 | 32.04 | 72.70 | 76.93 | 50.99 | 47.88 |
| Replay† | **83.04** | 83.48 | 59.22 | 62.27 | 85.90 | 93.36 | 56.52 | 62.45 |
| iCaRL† (Rebuffi et al., 2017) | 81.48 | 81.90 | 53.75 | 57.91 | 75.54 | 81.17 | 61.08 | 76.31 |
| MEMO† (Zhou et al., 2023) | 71.44 | 62.92 | 53.41 | 57.04 | 61.30 | 67.52 | 50.54 | 48.59 |
| SimpleCIL (Zhou et al., 2024a) | 68.45 | 75.72 | 38.18 | 44.08 | 71.63 | 74.80 | 63.19 | 63.40 |
| L2P (Wang et al., 2022d) | 75.30 | 79.03 | 45.15 | 50.90 | 83.63 | 87.57 | 67.01 | 70.56 |
| DualPrompt (Wang et al., 2022c) | 75.41 | 80.70 | 46.31 | 52.29 | 84.09 | 87.55 | 59.85 | 72.07 |
| CODA-Prompt (Smith et al., 2023) | 81.21 | 84.53 | 52.86 | 57.38 | 87.91 | 91.21 | 63.05 | 73.53 |
| EASE (Zhou et al., 2024b) | 75.96 | 74.76 | 47.31 | 44.62 | 86.32 | 86.69 | 64.08 | 69.19 |
| RanPAC (McDonnell et al., 2023) | 78.03 | 82.28 | 50.11 | 54.98 | 79.44 | 81.32 | 75.00 | 81.04 |
| S-iPrompt (Wang et al., 2022b) | 78.27 | 80.81 | 54.51 | 60.06 | 82.09 | 83.72 | 61.46 | 72.43 |
| DUCT | 81.95 | **86.90** | 62.04 | **68.17** | 91.70 | 93.99 | 80.72 | 84.91 |

## A  SUPPLIED EXPERIMENTAL RESULTS

In this section, we supply additional experiments to show the effectiveness of DUCT, including the running time comparison, the detailed performance among different task orders, and more results with other pre-trained weights.

Table 5: Average and last performance of different methods with the 3rd task order in Section B.2. The best performance is shown in bold. All methods are implemented with ViT-B/16 IN1K. Methods with † indicate implementations with exemplars (10 per class).

| Method | Office-Home | | DomainNet | | Core50 | | CDDB | |
|---|---|---|---|---|---|---|---|---|
| | $\bar{\mathcal{A}}$ | $\mathcal{A}_B$ | $\bar{\mathcal{A}}$ | $\mathcal{A}_B$ | $\bar{\mathcal{A}}$ | $\mathcal{A}_B$ | $\bar{\mathcal{A}}$ | $\mathcal{A}_B$ |
| Finetune | 81.82 | 74.93 | 44.84 | 29.45 | 77.35 | 79.12 | 51.68 | 52.81 |
| Replay† | 86.98 | 84.10 | 67.23 | 60.52 | 84.91 | 91.60 | 87.52 | 75.30 |
| iCaRL† (Rebuffi et al., 2017) | 84.40 | 81.64 | 61.74 | 54.41 | 75.54 | 81.17 | 88.66 | **86.05** |
| MEMO† (Zhou et al., 2023) | 75.50 | 63.43 | 62.97 | 56.99 | 68.42 | 71.78 | 53.94 | 52.55 |
| SimpleCIL (Zhou et al., 2024a) | 76.32 | 75.72 | 45.74 | 44.08 | 69.67 | 74.80 | 56.64 | 63.40 |
| L2P (Wang et al., 2022d) | 81.94 | 79.57 | 52.16 | 45.05 | 84.21 | 88.24 | 67.87 | 67.87 |
| DualPrompt (Wang et al., 2022c) | 82.45 | 80.79 | 54.01 | 49.28 | 84.91 | 88.39 | 66.26 | 71.19 |
| CODA-Prompt (Smith et al., 2023) | 63.05 | 73.53 | 60.91 | 56.08 | 88.58 | 91.20 | 68.77 | 76.26 |
| EASE (Zhou et al., 2024b) | 81.45 | 74.76 | 52.19 | 40.81 | 86.26 | 85.26 | 67.48 | 72.18 |
| RanPAC (McDonnell et al., 2023) | 83.84 | 82.28 | 57.57 | 54.08 | 78.07 | 81.62 | 78.46 | 81.04 |
| S-iPrompt (Wang et al., 2022b) | 83.40 | 80.68 | 64.02 | 61.22 | 82.65 | 84.20 | 69.30 | 72.99 |
| DUCT | **87.31** | **86.94** | **70.08** | **67.06** | **91.97** | **94.78** | **88.84** | 85.84 |

Table 6: Average and last performance of different methods with the 4th task order in Section B.2. The best performance is shown in bold. All methods are implemented with ViT-B/16 IN1K. Methods with † indicate implementations with exemplars (10 per class).

| Method | Office-Home | | DomainNet | | Core50 | | CDDB | |
|---|---|---|---|---|---|---|---|---|
| | $\bar{\mathcal{A}}$ | $\mathcal{A}_B$ | $\bar{\mathcal{A}}$ | $\mathcal{A}_B$ | $\bar{\mathcal{A}}$ | $\mathcal{A}_B$ | $\bar{\mathcal{A}}$ | $\mathcal{A}_B$ |
| Finetune | 82.10 | 77.09 | 25.82 | 16.65 | 75.97 | 75.21 | 53.80 | 50.42 |
| Replay† | 86.38 | 82.67 | **65.35** | 60.26 | 85.89 | 92.23 | 89.91 | 77.27 |
| iCaRL† (Rebuffi et al., 2017) | 83.38 | 78.71 | 58.30 | 51.55 | 68.25 | 72.91 | **92.67** | **88.60** |
| MEMO† (Zhou et al., 2023) | 71.84 | 65.06 | 62.28 | 54.18 | 68.20 | 69.76 | 87.53 | 80.93 |
| SimpleCIL (Zhou et al., 2024a) | 81.31 | 75.72 | 40.06 | 44.08 | 71.73 | 74.80 | 66.56 | 63.40 |
| L2P (Wang et al., 2022d) | 85.49 | 81.71 | 48.59 | 45.47 | 83.35 | 87.01 | 77.38 | 61.50 |
| DualPrompt (Wang et al., 2022c) | 84.74 | 81.02 | 50.90 | 46.77 | 85.23 | 86.89 | 81.46 | 72.89 |
| CODA-Prompt (Smith et al., 2023) | 71.06 | 74.84 | 60.17 | 55.89 | 88.10 | 91.79 | 79.73 | 73.98 |
| EASE (Zhou et al., 2024b) | 84.69 | 74.76 | 48.28 | 42.92 | 86.30 | 86.69 | 70.23 | 50.48 |
| RanPAC (McDonnell et al., 2023) | 86.59 | 82.28 | 54.97 | 53.48 | 79.31 | 81.32 | 85.28 | 79.66 |
| S-iPrompt (Wang et al., 2022b) | 85.52 | 80.19 | 61.32 | 60.89 | 82.11 | 83.50 | 81.28 | 73.06 |
| DUCT | **89.79** | **86.98** | 64.12 | **64.70** | **92.12** | **94.56** | 89.38 | 84.31 |

## A.1 RUNNING TIME COMPARISON

In this section, we report the running time comparison among different compared methods. As shown in Figure 7, DUCT costs competitive running time against other compared methods while having the best performance.

## A.2 PERFORMANCE OF DIFFERENT TASK ORDERS

In the main paper, we conduct experiments on the benchmark datasets with five task orders and report the average performance. In this section, we report the performance on each order in Table 3, 4, 5, 6, 7. The task sequences are reported in Section B.2.

## A.3 DIFFERENT BACKBONES

In the main paper, we mainly report the performance with ViT-B/16-IN1K. Since DUCT is robust with the change of backbones, we report the performance with ViT-B/16-IN21K in this section. Apart from the benchmark datasets reported in Figure 2, the performance on the rest of the benchmarks is shown in Figure 8.

## B DATASET DETAILS

In this section, we introduce the details about datasets, including the dataset information (*i.e.*, the number of tasks and instances) and split information (*i.e.*, the five splits adopted in the main paper).

Table 7: Average and last performance of different methods with the 5th task order in Section B.2. The best performance is shown in bold. All methods are implemented with ViT-B/16 IN1K. Methods with † indicate implemented with exemplars (10 per class).

| Method | Office-Home $\bar{\mathcal{A}}$ | $\mathcal{A}_B$ | DomainNet $\bar{\mathcal{A}}$ | $\mathcal{A}_B$ | Core50 $\bar{\mathcal{A}}$ | $\mathcal{A}_B$ | CDDB $\bar{\mathcal{A}}$ | $\mathcal{A}_B$ |
|---|---|---|---|---|---|---|---|---|
| Finetune | 77.69 | 78.18 | 38.49 | 26.9 | 76.73 | 77.52 | 53.40 | 49.21 |
| Replay† | 84.12 | 84.72 | 67.45 | 59.88 | 85.37 | 92.22 | 50.34 | 50.12 |
| iCaRL† (Rebuffi et al., 2017) | 80.99 | 81.26 | 62.11 | 54.14 | 74.86 | 77.55 | 50.10 | 49.77 |
| MEMO† (Zhou et al., 2023) | 66.95 | 64.24 | 70.29 | 63.49 | 61.18 | 64.61 | 53.36 | 50.35 |
| SimpleCIL (Zhou et al., 2024a) | 80.75 | 75.72 | 51.15 | 44.08 | 70.78 | 74.80 | 63.06 | 63.40 |
| L2P (Wang et al., 2022d) | 81.37 | 81.38 | 57.33 | 51.13 | 83.21 | 88.22 | 65.68 | 54.42 |
| DualPrompt (Wang et al., 2022c) | 82.55 | 81.00 | 57.33 | 48.12 | 82.99 | 83.48 | 71.00 | 69.18 |
| CODA-Prompt (Smith et al., 2023) | 63.33 | 72.28 | 66.85 | 57.41 | 87.33 | 92.81 | 71.06 | 74.84 |
| EASE (Zhou et al., 2024b) | 85.13 | 80.23 | 51.68 | 44.62 | 86.37 | 87.47 | 70.65 | 60.75 |
| RanPAC (McDonnell et al., 2023) | 84.31 | 82.28 | 61.48 | 54.98 | 79.52 | 81.32 | 83.39 | 79.64 |
| S-iPrompt (Wang et al., 2022b) | 82.70 | 80.47 | 67.73 | 61.23 | 81.98 | 83.38 | 68.76 | 72.09 |
| DUCT | **88.55** | **86.92** | **72.29** | **66.59** | **91.88** | **94.21** | **83.46** | **85.48** |

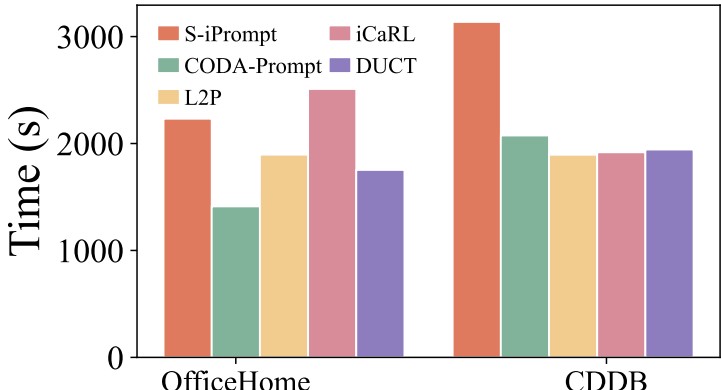

Figure 7: Running time comparison among different methods. DUCT shows the best performance while having competitive training costs.

## B.1 DATASET INTRODUCTION

We report the details about benchmark datasets in Table 8, and they are listed as follows.

- **DomainNet** (Peng et al., 2019)[1] is a dataset of common objects in six different domains. All domains include 345 categories of objects, such as Bracelets, planes, birds, and cellos. The domains include clipart — a collection of clipart images; real — photos and real-world images; sketch — sketches of specific objects; infograph — infographic images with specific objects; painting — artistic depictions of objects in the form of paintings, and quickdraw — drawings of the worldwide players of the game 'Quick Draw!'. We use the officially recommended version 'Cleaned' in this paper.

- **Office-Home** (Venkateswara et al., 2017)[2] is a benchmark dataset for domain adaptation which contains four domains where each domain consists of 65 categories. The four domains are Art — artistic images in the form of sketches, paintings, ornamentation, etc.; Clipart — a collection of clipart images; Product — images of objects without a background; and Real-World — images of objects captured with a regular camera. It contains 15,500 images, with an average of around 70 images per class and a maximum of 99 images in a class.

---

[1]https://ai.bu.edu/M3SDA/
[2]https://hemanthdv.github.io/officehome-dataset/

Table 8: Details on domain size, train/test split, and instance number of the benchmark datasets. The dataset split and selection follows (Wang et al., 2022c;d;b; Smith et al., 2023).

|  | Domains | Size | Test set |
|---|---|---|---|
| CDDB-Hard | biggan | 4.0k | standard splits 75%:25% ('san' - 80%:20%) |
|  | gaugan | 10.0k |  |
|  | san | 440 |  |
|  | whichfaceisreal | 2.0k |  |
|  | wild | 10.5k |  |
| CORe50 | s1 | 14.9k | s3, s7, s10 Indoor:Outdoor |
|  | s2 | 14.9k |  |
|  | s4 | 14.9k |  |
|  | s5 | 14.9k |  |
|  | s6 | 14.9k |  |
|  | s8 | 14.9k |  |
|  | s9 | 14.9k |  |
|  | s11 | 14.9k |  |
| DomainNet | clipart | 48.1k | standard splits 70%:30% |
|  | infograph | 51.6k |  |
|  | painting | 72.2k |  |
|  | quickdraw | 172.5k |  |
|  | real | 172.9k |  |
|  | sketch | 69.1k |  |
| Office-Home | Art | 2.4k | random splits 70%:30% |
|  | Clipart | 4.3k |  |
|  | Product | 4.4k |  |
|  | Real World | 4.3k |  |

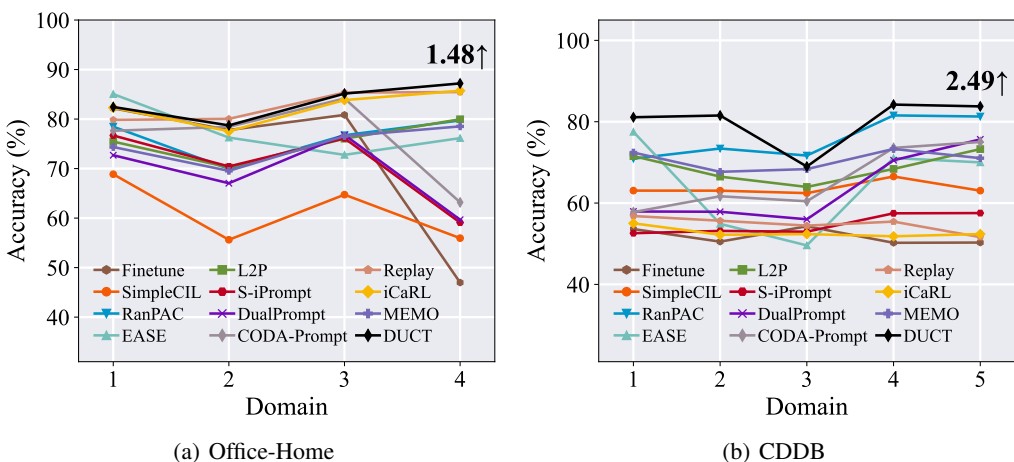

(a) Office-Home  (b) CDDB

Figure 8: Incremental performance of different methods with ViT-B/16 IN21K. We report the performance gap after the last incremental stage between DUCT and the runner-up method at the end of the line.

- **CDDB-Hard** (Li et al., 2023)[3]. As a dataset mixed with real-world and model-generated images, the continual deepfake detection benchmark (CDDB) aims to simulate real-world deepfakes' evolution. The authors put out three scenarios for evaluation, 'EASY,' 'Hard,'

---

[3]https://coral79.github.io/CDDB_web/

Table 9: Task orders of CDDB-Hard.

| CDDB-Hard | Task 1 | Task 2 | Task 3 | Task 4 | Task 5 |
|---|---|---|---|---|---|
| Order 1 | san | whichfaceisreal | biggan | wild | gaugan |
| Order 2 | wild | whichfaceisreal | san | gaugan | biggan |
| Order 3 | biggan | gaugan | wild | whichfaceisreal | san |
| Order 4 | gaugan | biggan | wild | whichfaceisreal | san |
| Order 5 | whichfaceisreal | san | gaugan | biggan | wild |

Table 10: Task orders of CORe50.

| CORe50 | Task 1 | Task 2 | Task 3 | Task 4 | Task 5 | Task 6 | Task 7 | Task 8 |
|---|---|---|---|---|---|---|---|---|
| Order 1 | s11 | s4 | s2 | s9 | s1 | s6 | s5 | s8 |
| Order 2 | s2 | s9 | s1 | s6 | s5 | s8 | s11 | s4 |
| Order 3 | s4 | s1 | s9 | s2 | s5 | s6 | s8 | s11 |
| Order 4 | s1 | s9 | s2 | s5 | s6 | s8 | s11 | s4 |
| Order 5 | s9 | s2 | s5 | s6 | s8 | s11 | s4 | s1 |

and 'Long,' respectively. Following (Wang et al., 2022b), we choose the 'Hard' track for evaluation since it poses a more challenging problem due to its complexity.

- **CORe50** (Lomonaco & Maltoni, 2017)[4]. Composed of 11 sessions characterized by different backgrounds and lighting, CORe50 is built for continual object recognition. Numerous RGB-D images are divided into 8 indoor sessions for training and 3 outdoor sessions for testing, and each session includes a sequence of about 300 frames for all 50 objects.

## B.2 DOMAIN SEQUENCES

In domain-incremental learning, different algorithms' performances may be influenced by the order of domains. Consequently, we randomly shuffle the domains and organize five domain orders in the main paper, which are further utilized for a holistic evaluation. The task orders are reported in Table 9, 10, 11, 12.

## C COMPARED METHODS

In this section, we introduce the methods that were compared in the main paper. **Note that we re-implement all methods using the same pre-trained model as initialization**. They are listed as follows.

- **Finetune** is a simple baseline in DIL, which directly optimizes the model with cross-entropy loss. It will suffer catastrophic forgetting since there is no restriction on preserving previous knowledge.

- **Replay** (Ratcliff, 1990) is an exemplar-based method, which saves a set of exemplars from previous domains (*i.e.*, in this paper, we save 10 exemplars per class) and replay them when learning new domains. Hence, forgetting can be alleviated since the model can revisit informative instances from previous domains when learning new ones. Of note, for classes with fewer than 10 instances, repeatable sampling is allowed to conform to the requirement.

- **iCaRL** (Rebuffi et al., 2017) is a knowledge distillation-based continual learning algorithm, which saves the previous model in memory. During updating, apart from the cross-entropy loss for learning new tasks, it also introduces the knowledge distillation loss between old and new models to avoid forgetting. It also requires saving an increasing number of exemplars.

- **MEMO** (Zhou et al., 2023) is an expansion-based continual learning algorithm that partially expands the network to catch new features. As for the implementation, we follow the original paper to decouple the network and expand the last transformer block for each new task.

---

[4]https://vlomonaco.github.io/core50/index.html#dataset

Table 11: Task orders of DomainNet.

| DomainNet | Task 1 | Task 2 | Task 3 | Task 4 | Task 5 | Task 6 |
|---|---|---|---|---|---|---|
| Order 1 | clipart | infograph | painting | quickdraw | real | sketch |
| Order 2 | infograph | painting | quickdraw | real | sketch | clipart |
| Order 3 | painting | quickdraw | real | sketch | clipart | infograph |
| Order 4 | quickdraw | real | sketch | clipart | infograph | painting |
| Order 5 | real | quickdraw | painting | sketch | infograph | clipart |

Table 12: Task orders of Office-Home.

| Office-Home | Task 1 | Task 2 | Task 3 | Task 4 |
|---|---|---|---|---|
| Order 1 | Art | Clipart | Product | Real_World |
| Order 2 | Clipart | Product | Real_World | Art |
| Order 3 | Product | Clipart | Real_World | Art |
| Order 4 | Real_World | Product | Clipart | Art |
| Order 5 | Art | Real_World | Product | Clipart |

- **SimpleCIL** (Zhou et al., 2024a) proposes this simple baseline in pre-trained model-based continual learning. It freezes the backbone representation, extracts the class center of each class, and utilizes a cosine classifier updated by assigning class centers to the classifier weights.

- **L2P** (Wang et al., 2022d) is the first work introducing prompt tuning in continual learning. With the pre-trained weights frozen, it learns a prompt pool containing many prompts. During training and inference, instance-specific prompts are selected to produce the instance-specific embeddings. However, as alluded to before, learning new domains will lead to the overwriting of existing prompts, thus triggering forgetting.

- **DualPrompt** (Wang et al., 2022c) extends L2P in two aspects. Apart from the prompt pool and prompt selection mechanism, it further introduces prompts instilled at different depths and task-specific prompts. During training and inference, the instance-specific and task-specific prompts work together to adjust the embeddings.

- **CODA-Prompt** (Smith et al., 2023) aims to avoid the prompt selection cost in L2P. It treats prompts in the prompt pool as bases and utilizes the attention results to combine multiple prompts as the instance-specific prompt.

- **EASE** (Zhou et al., 2024b) designs lightweight feature expansion technique with adapters to learn new features as data devolves. To fetch a classifier with the same dimension as ever-expanding features, it utilizes class-wise similarity to complete missing class prototypes.

- **RanPAC** (McDonnell et al., 2023) extends SimpleCIL by randomly projecting the features into the high-dimensional space and learning the online LDA classifier for final classification.

- **S-iPrompt** (Wang et al., 2022b) is specially designed for pre-trained model-based domain-incremental learning. It learns task-specific prompts for each domain and saves domain centers in the memory with K-Means. During inference, it first forwards the features to select the nearest domain center via KNN search. Afterward, the selected prompt will be appended to the input.

