# OpenReview forum: "Dual Consolidation for Pre-Trained Model-Based Domain-Incremental Learning"
_ICLR.cc/2025/Conference — ICLR 2025 Conference Withdrawn Submission_

### Official Review · Reviewer_Y8cX · 2024-11-01

**Soundness:** 3
**Presentation:** 2
**Contribution:** 2
**Rating:** 5
**Confidence:** 4

**Summary:**

The paper addresses the problem of exemplar-free Domain-Incremental Learning (DIL) given a pre-trained embedding model. The proposed method, DUCT, jointly consolidate historical knowledge at both the representation level and the classifier level. At the representation level, the authors modify the technique of task vectors via considering the task similarity. At the classifier level, the authors propose to retrain a new classifier, and leverage the new classifier to modify the old classifier via optimal transport. To evaluate its effectiveness, DUCT is compared with DIL baselines on four cross-domain benchmarks. DUCT achieves state-of-the-art performance on all the experiments. An ablation study as well as other analytical experiments are reported to provide a more in-depth analysis of the proposed method.

**Strengths:**

1. The problem of exemplar-free domain-incremental learning is more challenging yet more practical. The authors did a good job maintaining the historical knowledge without replaying the past data.
2. The proposed method has strong performance, achieving a significant accuracy improvement compared to existing DIL methods.
3. The algorithm is simple, which could make a broader impact to the community.

**Weaknesses:**

1. The major concern is that, as shown in the ablation study in table 2, the main reason for the accuracy boost in DUCT could be attributed to task vector, which is an existing technique for addressing multiple tasks simultaneously. In reviewer's opinion, despite that the authors make certain modification on the weighting strategy, the paper fails to provide new insights to this technique on why it is effective in addressing DIL problem. One possible aspect the reviewer can think of is to explain why applying DUCT 'places the same class of different domains together', as suggested in the visualization in fig. 5.

2. The notation in the paper needs improvement. First, in equation 5, $\phi^m_i$ should not use $i$ as subscript as it indicates the index of the summation. Second, $\beta$ and $\gamma$ should be explained once they appear in line 300.

**Questions:**

1. As shown in fig. 2, the initial performance of DUCT on the first domain is not optimal. Please further elaborate on this issue.

2. The parameter sensitivity analysis in fig. 3(c) indicates that DUCT still achieves decent performance when the head-merge ratio $\alpha_W$ is small. What if the ratio is set to zero?

3. Can the proposed method be applied to class-incremental learning, given that it treats classes from the incoming domain as new categories?

---

### Official Review · Reviewer_ebkv · 2024-11-01

**Soundness:** 2
**Presentation:** 2
**Contribution:** 2
**Rating:** 5
**Confidence:** 4

**Summary:**

This paper is motivated by the forgetting problem of features and the mismatch problem of the classifier in domain-incremental learning. Then this paper proposes to address the above problems by unifying the historical knowledge at both the feature and classifier level. In particular, this paper proposes to merge the backbone of different stages and utilize optimal transport to adapt the classifier of old domains to both the new domain and the merged backbone. This paper conducts multiple experiments to demonstrate the effectiveness of the proposed method.

**Strengths:**

1.This paper is well-motivated. It is easy to understand the motivation of the method. It is reasonable to consider both feature extraction and classification in incremental learning.
2.The experimental results show that the proposed method performs better than the previous methods, which successfully demonstrates the effectiveness of the proposed method.

**Weaknesses:**

1.The contributions are slightly limited. Integrating the models to balance different stages of tasks has been applied in incremental learning, such as [1].
2.Many descriptions and notations are confusing. For example, in line 143 of page 3, a lack of explanation of “b” in “b|Y|”. In lines 179-185, the author seems to have used different words (features, representation, and embedding) to convey the same meaning, and I don't quite understand why the author did so. In Eq.(4), there lack of interpretation of , maxima value, and initial values in the summation notation. In line 257 of page 5, the description of “at most two backbones in memory” is confusing since I find there are at least three models (,, ) in memory according to lines 256. Finally, the authors are conflicted on which way to integrate, as shown in lines 256 and Algorithm 1.
[1] Zheng Z, Ma M, Wang K, et al. Preventing zero-shot transfer degradation in continual learning of vision-language models[C]//Proceedings of the IEEE/CVF International Conference on Computer Vision. 2023: 19125-19136.

**Questions:**

1.What is the impact on the results if the proposed method does not use task vectors during integration?
2.In lines 198-200 of page 4, the authors claim that the proposed method can capture the domain-specific features of all domains. This is an interesting claim. How do the authors prove it?

---

### Official Review · Reviewer_EjWt · 2024-11-02

**Soundness:** 3
**Presentation:** 3
**Contribution:** 3
**Rating:** 6
**Confidence:** 4

**Summary:**

I authors propose DUCT, a method for domain incremental learning (DIL). DIL is the setting where a sequence of tasks is presented during model finetuning. The training algorithm does not have access to data from prior tasks. The authors decompose the task overfitting problem into two components: (1) representation overfitting and (2) classifier overfitting. The authors tackle to two problems separately and propose novel model-merging-inspired techniques to solve both.

**Strengths:**

(1) The manuscript is well-organized despite the complicate method.

(2) The method is novel.

(3) Results are good.

**Weaknesses:**

(1) Many notations are not introduced before hand. This makes the math hard to follow and the method ambiguous. For example, what is $\phi_i^m$ and $\alpha_\phi$ in equation 4? Furthermore, equations are not introduced in the correct order. For example, Eq. 5 depends on a value that is not defined until Eq. 7.

(2) It is unclear why the proposed method is better than model merging (intuitively).

**Questions:**

Why do you need the two stage merging? Could you just absorb the linear weights into $\phi$ and use equation (5)?

Is there anyway you could calculate an upper bound for Table1 (e.g. performance of finetuning on the union of all datasets?)

---

### Official Review · Reviewer_f6C1 · 2024-11-05

**Soundness:** 2
**Presentation:** 3
**Contribution:** 2
**Rating:** 3
**Confidence:** 4

**Summary:**

The DUal ConsolidaTion (DUCT) framework addresses feature drift and forgetting by integrating historical backbones through a representation merging technique, creating stable task-specific embeddings. Additionally, DUCT’s classifier consolidation process merges calibrated and historical weights, preserving semantic class relationships and resisting forgetting, yielding strong performance on diverse benchmarks.

**Strengths:**

The authors analyze the current challenge of forgetting in domain incremental learning (DIL) and its underlying causes.
They propose a model-merging approach that demonstrates promising accuracy.

**Weaknesses:**

1. Equations 4 and 5 attempt to build a unified embedding through weighted summation of model weights, raising questions about feasibility. Given the complexity and lack of interpretability in deep network weights, is this combination effective, or could it intensify conflicts within the feature space? More comprehensive theoretical analysis is required.
2. While the authors suggest that DIL could benefit applications like autonomous vehicles and face recognition, their experiments focus on classification tasks. Testing on more realistic applications could be more convincing.
3. The proposed DUCT method relies on model merging. However, as domains accumulate, previously merged models may become overly complex, containing information from multiple domains, while models from newer domains include only the latest domain data. This could lead to an imbalance between older and newer domains, creating potential confusion and forgetting.
4. The authors tested DUCT on ViT-B/16, but other methods, like S-Prompts, report results on the more powerful CLIP backbone. Does DUCT maintain its effectiveness on a stronger backbone?

**Questions:**

see the weakness

---

### Official Review · Reviewer_wWjP · 2024-11-07

**Soundness:** 3
**Presentation:** 3
**Contribution:** 3
**Rating:** 6
**Confidence:** 4

**Summary:**

This paper introduces DUCT, a dual consolidation technique for domain-incremental learning (DIL) that effectively mitigates catastrophic forgetting. DUCT addresses the challenge of balancing knowledge across domains by Representation Consolidation and Classifier Consolidation. The paper demonstrates DUCT’s effectiveness through extensive experiments on four benchmark datasets, showing it consistently outperforms state-of-the-art methods in terms of accuracy and forgetting measure.

**Strengths:**

1.	DUCT introduces approach to domain-incremental learning by addressing both feature and classifier forgetting simultaneously, providing a fresh perspective on solving the catastrophic forgetting problem.
2.	DUCT cleverly combines ideas from model merging and optimal transport. The merging of task vectors with the pre-trained model and the use of optimal transport for classifier alignment are creative applications of existing techniques to the DIL context.
3.	The paper presents comprehensive experimental results across four benchmark datasets and five task orders, demonstrating the robustness and effectiveness of DUCT.
4.	This paper is well-organized, with clear sections and logical flow. The methodology is explained in detail, and the experimental setup and results are presented clearly.

**Weaknesses:**

1.	There are many vision language models like CLIP that can perform zero shot. Can the author report the results of CLIP and CLIP related fine-tuning methods such as Coop, CoCoop, etc., to demonstrate the advantages of the article's method compared to these general models.
2.	I wonder what data is used to calculate the class center of the pretrained model? If using pretrained data such as ImageNet, the first issue is how to ensure consistency with downstream task categories to calculate class center similarity? The second question is how to reduce the overhead caused by large number of categories and data size?
3.	The author should conduct experiments on more backbones to demonstrate the effectiveness of the method, such as convolutional neural networks like Resnet.
4.	Task similarity is calculated based on all categories, may it lead to the influence of some categories being overly magnified while the influence of others is ignored.

**Questions:**

See weakness.

---

### Note · Authors · 2024-11-14

I have read and agree with the venue's withdrawal policy on behalf of myself and my co-authors.